# LoRA in the Right Place: Which Block to Tune in Parameter-Efficient Fine-Tuning?

## Abstract

Are all blocks equally important in parameter-efficient fine-tuning? This fundamental question underlies almost every PEFT method, yet decisions about where to insert tunable parameters are often based on convention or ad hoc heuristics. In this work, we revisit this design decision by exploring the theoretical ground behind this choice, with the goal of developing a rigorous understanding of block-level placement within the PEFT paradigm. Starting from a simple scalar example, we show how perturbations in smaller blocks can be amplified through interactions with larger ones, and then extend this reasoning to matrices using norm-based analysis. Our results further reveal that the softmax operation tends to suppress updates to queries and keys, suggesting that value and output blocks should be prioritized. For tasks that rely on class tokens, we find that tuning the output block often outperforms the traditional emphasis on the value block. Importantly, this block-selection principle generalizes beyond the standard LoRA to other PEFT variants such as DoRA and AdaLoRA, underscoring its broad applicability. We validate these insights with extensive experiments across architectures, pretrained models, rank settings, and downstream benchmarks. Overall, our findings establish block selection as a key factor in PEFT and offer principled, empirically grounded strategies for improving both efficiency and effectiveness in model adaptation.

## 1 Introduction

Are all blocks equally important in parameter-efficient fine-tuning (PEFT) (Houlsby et al., 2019)? The choice of which blocks to adapt is not a minor technical detail; it directly determines how effectively a pretrained model can transfer to new tasks, how much compute and memory are required during fine-tuning, and ultimately how far parameter efficiency can be pushed in practice. Since PEFT methods update only a small subset of parameters, deciding which subset to tune becomes especially critical: an informed choice can unlock strong performance with minimal overhead, while a poor one can squander both efficiency and accuracy (Guo et al., 2021).

Historically, the design of PEFT methods often follows a set of simple but effective heuristics. When tuning only a single block, practitioners typically target the value projection, and when extending to two blocks, the conventional choice has been the query–value pair. These patterns originated from early empirical studies in LoRA (Hu et al., 2022) and have since become the default in most implementations. Subsequent works have introduced more sophisticated mechanisms, such as adaptive rank allocation (Zhang et al., 2023), layer sampling (Pan et al., 2024), and parameter pruning (Guo et al., 2021), to optimize the distribution of trainable resources. Despite these advances, the fundamental question of which blocks within each layer are intrinsically most important remains largely unexplored. In particular, rigorous analysis of why tuning certain blocks contribute more effectively to adaptation than others remains lacking, leaving an important gap in both theory and practice.

This paper takes a closer look at this question, aiming to uncover the principles behind the varying importance of different blocks in fine-tuning. Specifically, we begin with a toy example that illuminates the dynamics of tuning small-scale blocks, showing how their perturbations can be amplified by larger ones. Building on these insights, we extend our analysis to full layers, systematically evaluating the importance of each block in modern transformer architectures (Vaswani et al., 2017; Dosovitskiy et al., 2021).

Our analysis leads to several key insights for block selection. We find that when the value and output blocks have comparable matrix norms, the output block should be prioritized, especially in class-token–dependent tasks such as classification. When their norms differ markedly, tuning the block with the smaller norm is more effective, since its perturbations are naturally amplified by interactions with larger components. Our analysis also reveals that tuning the query and key blocks is often less effective, due to the dampening effect of the softmax operation. Finally, when two blocks can be tuned simultaneously, selecting the output and value blocks consistently reduces loss more effectively than the conventional query–value pairing in LoRA (Hu et al., 2022).

To validate our block-selection insights, we conduct extensive experiments across diverse architectures, pretrained models, rank configurations, and downstream tasks. We systematically compare tuning individual blocks as well as block combinations, tracking both training dynamics and final performance. These experiments validate our theoretical principles—such as the amplified effect of smaller-norm blocks, the central role of the output block, and the limited impact of query/key tuning—hold in practice. For instance, in image classification tasks, we consistently observe that prioritizing the output block yields noticeable improvements in both training dynamics and final accuracy, particularly in low-rank settings. Importantly, this pattern is not restricted to a single PEFT method: comparable gains appear across frameworks including LoRA, AdaLoRA and DoRA (Liu et al., 2024). These results reinforce the generality of our block-selection principles and confirm that output-block prioritization is a general strategy across architectures and adaptation methods.

## 2 RELATED WORKS

**Parameter-Efficient Fine-tuning** The rise of large-scale pre-trained models has revolutionized the field of artificial intelligence in multiple areas (Devlin et al., 2019; Liu et al., 2019; Dosovitskiy et al., 2021; Gong et al., 2021; Chen et al., 2022a). Yet, the sheer size of these models makes fine-tuning them on downstream tasks computationally expensive and memory-intensive. To mitigate these challenges, parameter-efficient fine-tuning (PEFT) (Howard & Ruder, 2018; Houlsby et al., 2019) has emerged as a practical solution, enabling task adaptation without updating the full model. This paradigm encompasses several key categories. Adapter-based methods introduce small, new modules or adapters into the pre-trained model and only fine-tune these new parameters (Houlsby et al., 2019; He et al., 2022a; Zhou et al., 2024). In contrast, prompt- and prefix-tuning approaches freeze the entire model and instead optimize a small, continuous prompt that is prepended to the input sequence (Lester et al., 2021; Li & Liang, 2021). Another prominent category is low-rank adaptation (LoRA), which modifies the pre-trained model's existing weights by injecting low-rank matrices into the original weight matrices (Hu et al., 2022; Liu et al., 2024). Recent studies in Zhang et al. (2024) have also shown such a tuning strategy is closely related to the classical control approaches (Franklin et al., 2002).

**Parameter Selection and Rank Allocation** While PEFT methods significantly reduce the number of trainable parameters, determining which components to adapt and how to allocate the parameter budget remains a critical challenge. Early work explored layer-wise adaptation strategies, with findings suggesting that fine-tuning later layers is more effective for downstream tasks (Kenton et al., 2018; Peters et al., 2019). Recent advances have focused on adaptive selection mechanisms, such as AdaLoRA (Zhang et al., 2023) to dynamically allocate rank budgets, LISA (Pan et al., 2024) to sample layers, DiffPruning (Guo et al., 2021) to utilize gradient-based importance measures, Jin et al. (2023) to use smaller models, and Zangrando et al. (2025) to consider bilevel optimization. Furthermore, sparse selection methods (Ansell et al., 2022) and mixture-of-adapters approaches (Wang et al., 2022) have demonstrated that strategic parameter selection can achieve comparable performance to full-model adaptation while maintaining computational efficiency. These selection strategies have proven particularly valuable in multi-task scenarios (Pfeiffer et al., 2020; Üstün et al., 2020), where different tasks may benefit from adapting different model components. Overall, our work differs from these adaptive strategies by focusing on the intrinsic importance of individual blocks within each layer, and by employing a fixed block-selection strategy guided by both theoretical and empirical analysis. Similar to LoRA, our approach is also data-agnostic, therefore allowing simple implementations in practice. Meanwhile, our method is *complementary and compatible* with existing adaptive mechanisms: for example, one could first identify the most critical blocks and then apply dynamic rank allocation as AdaLoRA (Zhang et al., 2023), or first select layers as in LISA (Pan et al., 2024) and subsequently choose the optimal blocks within them.

# 3 BLOCKRANK: BLOCK-LEVEL SELECTION FOR PEFT

In this section, we first formally define the problem of block selection in the context of low-rank adaptation and other PEFT paradigms. Based on this formulation, we then analyze the relative importance of individual blocks within each layer, offering theoretical insights to guide more effective and efficient fine-tuning strategies.

## 3.1 PROBLEM FORMULATION

Consider a pre-trained transformer model with $L$ layers, where each layer consists of multiple blocks, including attention projections (query, key, value, output). Let $\mathcal{B}_l = \{b_1, b_2, \ldots, b_{n_l}\}$ denote the set of blocks in layer $l$, where $n_l$ is the number of blocks in that layer.

In parameter-efficient fine-tuning (PEFT), the goal is to update only a small subset of parameters while keeping the majority of pre-trained weights frozen. For example, for a given block $b \in \mathcal{B}_l$ with weight matrix $W_b \in \mathbb{R}^{d_{\text{in}} \times d_{\text{out}}}$, a low-rank adaptation (LoRA) is introduced as

$$\Delta W_b = A_b B_b^\top, \quad A_b \in \mathbb{R}^{d_{\text{in}} \times r}, \quad B_b \in \mathbb{R}^{d_{\text{out}} \times r}, \quad r \ll \min(d_{\text{in}}, d_{\text{out}}).$$

Overall, block-level PEFT aims to identify a small subset of blocks in each layer that maximizes downstream task performance:

$$\min_{\mathcal{S}_1, \ldots, \mathcal{S}_L} \quad \mathcal{L}\Big(\theta + \Delta\theta(\{\mathcal{S}_l\}_{l=1}^L)\Big)$$

$$\text{s.t.} \quad \mathcal{S}_l \subseteq \mathcal{B}_l, \quad |\mathcal{S}_l| \leq k_l, \quad \forall l = 1, \ldots, L,$$

where $\theta$ denotes the pre-trained parameters, $k_l$ is the maximum number of tunable blocks allowed in layer $l$. For example, in the LoRA case, the total update from the selected blocks is

$$\Delta\theta\big(\{\mathcal{S}_l\}_{l=1}^L\big) = \bigcup_{l=1}^L \bigcup_{b \in \mathcal{S}_l} \Delta W_b.$$

For clarity, we restrict our study to the single-modality setting and focus on tuning blocks within the attention modules as the original LoRA work. The attention blocks (query, key, value, and output) share consistent dimensions across layers for $\forall l \in \{1, 2, \cdots, L\}$, ensuring that low-rank updates introduce the same number of trainable parameters. In contrast, MLP layers often involve substantial dimension changes; for instance, the hidden dimension in the first MLP layer is typically four times larger than its input dimension.

## 3.2 BLOCK RANKING FOR PEFT

**TL;DR.** *In attention layers with comparable projection norms, the sensitivity hierarchy is $W_O \geqslant W_V \gg W_Q \approx W_K$. If $W_O$ and $W_V$ differ markedly in norms, prioritize tuning the smaller-norm matrix.*

### 3.2.1 A TOY EXAMPLE

Directly analyzing block-level PEFT in full transformers is nevertheless challenging due to the complexity of interactions among layers and blocks. To build intuition, let us start with a toy example.

Suppose the output $y$ is a positive scalar function of two positive parameters $\theta_1, \theta_2 > 0$:

$$y = \theta_1 \theta_2 x,$$

where $x > 0$ is a fixed input. Suppose we are allowed to update only one parameter $\theta_i$ with a small change $0 < \delta\theta_i \ll \theta_i$, for $i \in \{1, 2\}$, and our goal is to reduce $y$ toward the target value 0. Then the following proposition illustrates the proper ranking for these two weight scalars.

**Proposition 1** (Optimal Parameter to Decrease Output). *Let $\theta_1, \theta_2, x > 0$, and $0 < \delta\theta_i \ll \theta_i$. If $0 < \theta_1 < \theta_2$, then updating $\theta_1$ yields a larger reduction in $y$:*

$$\theta_2(\theta_1 - \delta\theta_1)x - \theta_1\theta_2 x \leq \theta_1(\theta_2 - \delta\theta_2)x - \theta_1\theta_2 x,$$

*for any admissible $\delta\theta_i$.*

This toy example illustrates the intuition behind block-level PEFT: when the objective is to reduce the output, tuning parameters or blocks with smaller magnitudes is often more effective. A small adjustment to the smaller parameter is effectively *amplified* through its interaction with the larger one, producing a proportionally greater influence on the output.

### 3.2.2 THE MULTI-DIMENSIONAL LINEAR CASE

This amplification effect is not limited to scalar parameters; it also occurs in the matrix setting. Consider two matrices $W_1 \in \mathbb{R}^{m \times p}$ and $W_2 \in \mathbb{R}^{p \times n}$, with their product $W = W_1 W_2$. The following theorem formalizes how perturbations propagate in such products.

**Proposition 2** (Sensitivity in a matrix product). *Let $W_1 \in \mathbb{R}^{m \times p}$, $W_2 \in \mathbb{R}^{p \times n}$, and $W = W_1 W_2$. For any perturbations $\Delta W_1, \Delta W_2$,*

$$\frac{\|\Delta W_1\, W_2\|_F}{\|\Delta W_1\|_F} \leq \|W_2\|_2, \qquad \frac{\|W_1\, \Delta W_2\|_F}{\|\Delta W_2\|_F} \leq \|W_1\|_2.$$

Proposition 2 indicates that the magnitude of a perturbation in one matrix is constrained by the norm of the other matrix. As a result, changes applied to the matrix with smaller norm can have an outsized impact on the product, potentially producing the largest possible effect. This generalizes the intuition from the scalar example: tuning smaller blocks can lead to disproportionately large changes when they interact with larger blocks, highlighting their potential importance in block-level PEFT.

### 3.2.3 ATTENTION LAYERS FOR TRANSFORMERS

But the practical attention layer in modern transformers is not a simple composition of four independent matrix multiplications. Instead, it defines a more complex function due to the inclusion of the softmax and interactions between the query, key, and value projections. Specifically, we can formulate the function computed by attention as

$$F(X, W_Q, W_K, W_V, W_O) := \mathrm{softmax}\left(\frac{XW_Q^\top W_K X^\top}{\sqrt{d}}\right) XW_V W_O, \tag{1}$$

where $X \in \mathbb{R}^{n \times d}$ is the input, and $W_Q, W_K, W_V, W_O \in \mathbb{R}^{d \times d}$ are the query, key, value, and output projection matrices, respectively.

**Theorem 3** (Sensitivity Bounds for Attention). *Let $X \in \mathbb{R}^{n \times d}$ have unit-norm rows, and define*

$$S = \frac{XW_Q^\top W_K X^\top}{\sqrt{d}}, \qquad A = \mathrm{softmax}(S). \tag{2}$$

*For the $i$-th row of $S$, denote $s_i^\top$ and define its logit margin as*

$$\gamma_i := \max_j s_{i,j} - \max_{j \neq \arg\max s_{i,\cdot}} s_{i,j} \,(\geq 0), \qquad \gamma_{\min} := \min_{1 \leq i \leq n} \gamma_i.$$

*Then, for any perturbations $\Delta W_Q, \Delta W_K, \Delta W_V, \Delta W_O$, the following bounds hold:*

$$\frac{\|D_{W_Q}F[\Delta W_Q]\|_F}{\|\Delta W_Q\|_F} \leq \frac{2\min\{(n-1)e^{-\gamma_{\min}}, 1\}}{\sqrt{d}} \, \|W_O\|_2 \|W_V\|_2 \|W_K\|_2 \|X\|_2^3, \tag{3}$$

$$\frac{\|D_{W_K}F[\Delta W_K]\|_F}{\|\Delta W_K\|_F} \leq \frac{2\min\{(n-1)e^{-\gamma_{\min}}, 1\}}{\sqrt{d}} \, \|W_O\|_2 \|W_V\|_2 \|W_Q\|_2 \|X\|_2^3, \tag{4}$$

$$\frac{\|D_{W_V}F[\Delta W_V]\|_F}{\|\Delta W_V\|_F} \leq \|W_O\|_2 \|A\|_2 \|X\|_2, \tag{5}$$

$$\frac{\|D_{W_O}F[\Delta W_O]\|_F}{\|\Delta W_O\|_F} \leq \|A\|_2 \|W_V\|_2 \|X\|_2. \tag{6}$$

The bounds in Theorem 3 reveal a clear hierarchy in the sensitivity of attention blocks. The $W_Q$ and $W_K$ pathways include an additional factor $\min\{(n-1)e^{-\gamma_{\min}}, 1\}$, which decays *exponentially* with the minimum row-wise logit margin $\gamma_{\min}$. We have the following quantitative results on the comparison between the sensitivity estimation on the weights.

**Theorem 4.** *Let $X \in \mathbb{R}^{n \times d}$ have i.i.d. rows $x_i^\top$ drawn uniformly from the unit sphere $\mathbb{S}^{d-1}$. Write $M := W_Q^\top W_K$ and $M_{\mathrm{sym}} := (M + M^\top)/2$. Assume the weight scales are comparable:*

$$c \,\leq\, \|W_Q\|_2, \|W_K\|_2, \|W_V\|_2, \|W_O\|_2 \,\leq\, \tau c \qquad (\textit{some } c > 0,\ \tau \geq 1),$$

*and assume that*

$$\lambda_{\min}(M_{\mathrm{sym}}) \ge \alpha c^2, \qquad \|M\|_2 \le \beta c^2 \quad (\alpha \in (0,1],\ \beta \ge 1).$$

*Fix a failure probability $\delta \in (0, 1/2)$ and a target ratio $\eta \in (0,1)$. Set*

$$t_\delta := \sqrt{\frac{2}{d-1} \log \frac{2n(n-1)}{\delta}}, \qquad \chi_\delta := \sqrt{1 + (n-1)\, t_\delta}, \qquad a := \frac{\alpha - \beta t_\delta}{\sqrt{d}} \ (> 0).$$

*For $\delta \in (0,1)$, suppose*

$$c^2 \ \ge \ \frac{\sqrt{d}}{\alpha - \beta t_\delta} \max\Big\{ \log\big(2(n-1)\big),\, -W_{-1}\big(-K\big) \Big\} \quad \text{with} \quad K := \frac{\eta\, a\, \sqrt{d}}{2(n-1)\, \tau^2\, \chi_\delta^2}\ , ,\quad (7)$$

*where $W_{-1}$ is the $(-1)$ branch of the Lambert W function (Corless et al., 1996), i.e.,*

$$W_{-1}(x) \text{ is the unique real solution } w \text{ of } x = we^w \text{ with } w \le -1, \quad x \in \big[-\tfrac{1}{e}, 0\big). \qquad (8)$$

*Then, with probability at least $1 - \delta$ (over the draw of $X$), we have that the sensitivity upper bound in Theorem 2 for $W_Q$ and $W_K$ is at most an $\eta$-fraction of the sensitivity lower bound for $W_V$ and $W_O$.*

As a canonical example, the ViT-B/16 model typically uses $n = 197$, $d = 768$, and $\delta = 0.1$ (90% success). If we further assume $\alpha = 0.5$, and $\beta = 2$, and $\tau = 2$, then

$$t_{0.1} = \sqrt{\tfrac{2}{767} \log \tfrac{2 \cdot 197 \cdot 196}{0.1}} \approx 0.188, \quad \chi_{0.1} = \sqrt{1 + 196\, t_{0.1}} \approx 6.15, \quad a = \frac{1 - 0.188}{\sqrt{768}} \approx 0.0293.$$

For a half-factor dominance ($\eta = 1/2$), $K = \frac{\eta a}{2(n-1)\tau^2 \chi_\delta^2} \approx 6.8 \times 10^{-6}$, and $-W_{-1}(-K) \approx 10.97$, while $\log(2(n-1)) \approx 6.0$. Hence once

$$c^2 \ \ge \ 2452, \text{ i.e. } c \ge 49.51,$$

then with probability at least 90% (over the random draw of $X$) the first-order sensitivity upper bound of $F$ to $W_Q$ or $W_K$ is at most *half* that to $W_V$ or $W_O$ under same-size perturbations.

Consequently, when the attention distribution is sharp, small perturbations in $W_Q$ or $W_K$ have a strongly diminished effect on the output. In contrast, the $W_V$ and $W_O$ pathways are not subject to exponential damping, so updates to these blocks propagate more directly and can induce larger changes. Their effect is still upper-bounded by the spectral norm of the other matrix in the product—$\|W_O\|_2$ for $W_V$ and $\|W_V\|_2$ for $W_O$—similar to the linear matrix case. This analysis, consistent with our earlier scalar and matrix toy examples, suggests that in PEFT we should prioritize tuning the smaller module among $W_V$ and $W_O$ blocks.

### 3.2.4 $W_V$ VS. $W_O$: WHICH ONE FIRST?

For equal perturbation norms, Theorem 3 shows $\|D_{W_V} F\| \propto \|W_O\|_2$ and $\|D_{W_O} F\| \propto \|W_V\|_2$. However, there is also a directional controllability difference at the token level on tuning $W_O$ and $W_V$, especially when $W_O$ and $W_V$ are low-rank, which is usually the case in one head in the multi-head attention mechanism.

**Theorem 5.** *Let $b_i^\top$ denote the $i$-th row of $B := AX \in \mathbb{R}^{n \times d}$, where $A$ denotes the attention output as Theorem 3. Then for the $i$-th token output row $F_i^\top = b_i^\top W_V W_O$:*

- *(Perturb $W_V$) For any $\Delta W_V$, the first-order change is $\Delta F_i^\top = b_i^\top \Delta W_V W_O$. As $\Delta W_V$ varies arbitrarily, $\Delta F_i^\top$ is restricted to the row space of $W_O$, i.e. $\Delta F_i^\top \in \mathrm{row}(W_O) \subseteq \mathbb{R}^{1 \times d}$.*

- *(Perturb $W_O$) For any target $\Delta y \in \mathbb{R}^{1 \times d}$, if $b_i^\top W_V \ne 0$, there exists a $\Delta W_O$ such that $\Delta F_i^\top = b_i^\top W_V \Delta W_O = \Delta y$.*

*Consequently, for a specific token $i$, changing $W_O$ can realize arbitrary output directions, while changing $W_V$ is restricted to $\mathrm{row}(W_O)$.*

The above theorem indicates that for tasks requiring fine-grained, token-specific control (e.g., the class token) adjusting $W_O$ provides higher per-token flexibility. In contrast, updates to $W_V$ are constrained to the row space of $W_O$, limiting the range of achievable output directions. Therefore, when the downstream task (e.g., classification problem) relies on one specific token, prioritizing $W_O$ over $W_V$ can be more effective in steering the model's output.

## 3.3 COMPARISON WITH CONVENTIONAL PEFT HEURISTICS

Historically, PEFT design has followed simple heuristics, with single-block tuning typically applied to the value projection, as suggested by early empirical studies (Section 7.1 in Hu et al. (2022)). In contrast, our work derives target modules from theoretical analysis, showing that tuning the value and output matrices can have greater impact. Moreover, for single-block tuning, we find that prioritizing the output projection often yields higher local sensitivity than the value matrix when their norms are similar. This differs from the conventional approach, which typically emphasizes the value projection based on empirical heuristics. For the two-block case, our later experiment also finds that jointly tuning the value and output projections consistently achieves lower loss than the conventional query–value pair, demonstrating a principled improvement over heuristic-based designs.

## 4 EXPERIMENTS

We now present empirical evaluations of block selection in fine-tuning, examining how the choice of blocks affects convergence speed and final performance across different backbones, rank configurations, and datasets. Our analysis begins with the standard LoRA algorithm and then is extended to other PEFT methods to assess the consistency. Experimental setups are provided in Appendix D.

### 4.1 ViT EXPERIMENT

#### 4.1.1 SINGLE-BLOCK TUNING

We begin our experiments on the ViT model by fine-tuning a single block in each attention layer. Following prior work (Chen et al., 2022b), the model is evaluated on multiple image classification datasets to ensure consistent results. The backbone [1] is pretrained using the self-supervised Masked Autoencoder approach (He et al., 2022b). Since the classification task relies on a class token, our analysis in Theorem 5 suggests prioritizing the $O$ block when its scale is comparable to $V$, in contrast to the conventional LoRA setting, which typically favors the $V$ component. For completeness, we also examine the effects of tuning the $Q$ and $K$ blocks individually.

Table 1: Comparison of target blocks in LoRA fine-tuning on the self-supervised pretrained model with MAE. Trainable parameters include both the low-rank modules and the classification head.

| Configuration | Target | # of Parameters | CIFAR-100 | SVHN | Food-101 |
|---|---|---|---|---|---|
| LoRA (Rank-1) | Q | 0.10 M | $80.31_{\pm 0.21}$ | $93.50_{\pm 0.18}$ | $82.11_{\pm 0.21}$ |
| LoRA (Rank-1) | K | 0.10 M | $80.87_{\pm 0.26}$ | $93.32_{\pm 0.20}$ | $81.33_{\pm 0.39}$ |
| LoRA (Rank-1) | V | 0.10 M | $82.56_{\pm 0.14}$ | $94.89_{\pm 0.17}$ | $82.89_{\pm 0.17}$ |
| LoRA (Rank-1) | O | 0.10 M | $\mathbf{83.54}_{\pm 0.17}$ | $\mathbf{95.20}_{\pm 0.14}$ | $\mathbf{83.43}_{\pm 0.11}$ |
| LoRA (Rank-2) | Q | 0.11 M | $81.74_{\pm 0.16}$ | $94.58_{\pm 0.15}$ | $83.21_{\pm 0.19}$ |
| LoRA (Rank-2) | K | 0.11 M | $81.85_{\pm 0.19}$ | $94.38_{\pm 0.18}$ | $83.31_{\pm 0.19}$ |
| LoRA (Rank-2) | V | 0.11 M | $83.44_{\pm 0.17}$ | $95.71_{\pm 0.11}$ | $84.07_{\pm 0.11}$ |
| LoRA (Rank-2) | O | 0.11 M | $\mathbf{84.15}_{\pm 0.11}$ | $\mathbf{95.81}_{\pm 0.09}$ | $\mathbf{84.55}_{\pm 0.11}$ |
| LoRA (Rank-4) | Q | 0.15 M | $82.60_{\pm 0.14}$ | $95.36_{\pm 0.11}$ | $84.19_{\pm 0.21}$ |
| LoRA (Rank-4) | K | 0.15 M | $82.77_{\pm 0.16}$ | $95.43_{\pm 0.10}$ | $83.91_{\pm 0.18}$ |
| LoRA (Rank-4) | V | 0.15 M | $84.75_{\pm 0.07}$ | $96.42_{\pm 0.11}$ | $85.62_{\pm 0.13}$ |
| LoRA (Rank-4) | O | 0.15 M | $\mathbf{85.11}_{\pm 0.08}$ | $\mathbf{96.53}_{\pm 0.11}$ | $\mathbf{85.69}_{\pm 0.10}$ |
| LoRA (Rank-8) | Q | 0.22 M | $83.02_{\pm 0.16}$ | $95.93_{\pm 0.13}$ | $84.86_{\pm 0.19}$ |
| LoRA (Rank-8) | K | 0.22 M | $83.74_{\pm 0.17}$ | $95.89_{\pm 0.14}$ | $85.02_{\pm 0.14}$ |
| LoRA (Rank-8) | V | 0.22 M | $85.32_{\pm 0.08}$ | $96.81_{\pm 0.11}$ | $86.82_{\pm 0.14}$ |
| LoRA (Rank-8) | O | 0.22 M | $\mathbf{85.48}_{\pm 0.04}$ | $\mathbf{96.91}_{\pm 0.09}$ | $\mathbf{87.00}_{\pm 0.12}$ |

Table 1 summarizes the results of the ViT experiments across these three datasets. Across all ranks, fine-tuning the $O$ block consistently achieves the highest accuracy, particularly in low-rank settings. The $V$ block performs slightly lower, while tuning $Q$ or $K$ leads to noticeably worse results. These findings align with our theoretical analysis, which highlights the $O$ block as especially important for tasks dependent on the class token. For the rank-1 configuration, tuning $O$ outperforms $V$ by 0.98%, despite both strategies using the same number of trainable parameters. As the rank increases, the performance gap between $O$ and $V$ gradually narrows. Similar trends on SVHN and Food-101 indicate that the advantage of prioritizing the $O$ block generalizes across multiple datasets.

---

[1] https://github.com/facebookresearch/mae

### 4.1.2 SWITCHING TO A DIFFERENT PRETRAINED MODEL

To further validate our findings, we evaluate a different pretrained ViT-B model [2], which has been trained in a supervised manner on the ImageNet-21k dataset (Ridnik et al., 2021). This allows us to test whether the conclusions from our previous experiments, particularly the relative importance of tuning the $O$ and $V$ blocks, hold consistently across models with different initializations.

Table 2 shows the results of LoRA fine-tuning on the supervised pretrained ViT-B model. Consistent with the trends in Table 1, tuning the $O$ block achieves the highest accuracy across all ranks, followed closely by the $V$ block, while the $Q$ and $K$ blocks perform worse. The performance gap between $O$ and $V$ is slightly smaller than in the previous experiment, likely because the pretrained model already delivers strong results using only the classification head (e.g., $> 85\%$ on CIFAR-100). Nevertheless, the pattern across all datasets remains the same, confirming that the relative importance of the four attention blocks is consistent across different pretrained initializations. Additional results for two-block configurations are provided in Appendix F.

Table 2: Comparison of target blocks in LoRA fine-tuning on the supervised pretrained model. Trainable parameters include both the low-rank modules and the classification head.

| Configuration | Target | # of Parameters | CIFAR-100 | SVHN | Food-101 |
|---|---|---|---|---|---|
| LoRA (Rank-1) | Q | 0.10 M | $88.05_{\pm 0.17}$ | $90.87_{\pm 0.18}$ | $86.14_{\pm 0.18}$ |
| LoRA (Rank-1) | K | 0.10 M | $88.86_{\pm 0.17}$ | $91.25_{\pm 0.19}$ | $86.70_{\pm 0.16}$ |
| LoRA (Rank-1) | V | 0.10 M | $91.16_{\pm 0.14}$ | $94.32_{\pm 0.10}$ | $89.17_{\pm 0.13}$ |
| LoRA (Rank-1) | O | 0.10 M | $\mathbf{91.34}_{\pm 0.16}$ | $\mathbf{94.59}_{\pm 0.11}$ | $\mathbf{89.32}_{\pm 0.11}$ |
| LoRA (Rank-2) | Q | 0.11 M | $88.72_{\pm 0.18}$ | $92.50_{\pm 0.14}$ | $87.26_{\pm 0.20}$ |
| LoRA (Rank-2) | K | 0.11 M | $88.94_{\pm 0.09}$ | $92.78_{\pm 0.14}$ | $87.74_{\pm 0.21}$ |
| LoRA (Rank-2) | V | 0.11 M | $91.44_{\pm 0.10}$ | $95.32_{\pm 0.10}$ | $89.66_{\pm 0.11}$ |
| LoRA (Rank-2) | O | 0.11 M | $\mathbf{91.82}_{\pm 0.09}$ | $\mathbf{95.61}_{\pm 0.06}$ | $\mathbf{89.92}_{\pm 0.09}$ |
| LoRA (Rank-4) | Q | 0.15 M | $89.46_{\pm 0.12}$ | $93.62_{\pm 0.11}$ | $87.92_{\pm 0.17}$ |
| LoRA (Rank-4) | K | 0.15 M | $89.67_{\pm 0.11}$ | $93.97_{\pm 0.10}$ | $88.21_{\pm 0.17}$ |
| LoRA (Rank-4) | V | 0.15 M | $91.95_{\pm 0.06}$ | $96.04_{\pm 0.07}$ | $90.11_{\pm 0.13}$ |
| LoRA (Rank-4) | O | 0.15 M | $\mathbf{92.17}_{\pm 0.08}$ | $90.22_{\pm 0.06}$ | $\mathbf{90.27}_{\pm 0.05}$ |
| LoRA (Rank-8) | Q | 0.22 M | $90.00_{\pm 0.09}$ | $94.80_{\pm 0.10}$ | $88.54_{\pm 0.10}$ |
| LoRA (Rank-8) | K | 0.22 M | $90.14_{\pm 0.07}$ | $94.88_{\pm 0.09}$ | $88.49_{\pm 0.09}$ |
| LoRA (Rank-8) | V | 0.22 M | $92.00_{\pm 0.06}$ | $96.66_{\pm 0.06}$ | $90.52_{\pm 0.07}$ |
| LoRA (Rank-8) | O | 0.22 M | $\mathbf{92.23}_{\pm 0.04}$ | $\mathbf{96.74}_{\pm 0.06}$ | $\mathbf{90.56}_{\pm 0.06}$ |

### 4.1.3 EVALUATING THE IMPACT OF MATRIX NORMS WITH CONTROLLED MODIFICATIONS

In the above pretrained model, the spectral norms of the $O$ and $V$ blocks are generally comparable before fine-tuning. To investigate whether the relative size of the norms influences performance, we conducted *controlled modifications* of the pretrained weights. In one setup, we enlarged the $O$ block by three times while reducing $V$ by the same factor, ensuring that the final output remained unchanged; we refer to this configuration as "Large $O$". Figure 1 shows that in this case, fine-tuning the smaller $V$ block yields better training performance than tuning $O$. Conversely, we created a "Large $V$" setup by enlarging $V$ and shrinking $O$ by the same factor. Here, tuning the smaller $O$ block leads to superior training performance in subfigure (b). These findings indicate that when the scales of $O$ and $V$ differ, selecting the smaller block for fine-tuning can enhance performance, highlighting the importance of norm-aware block selection.

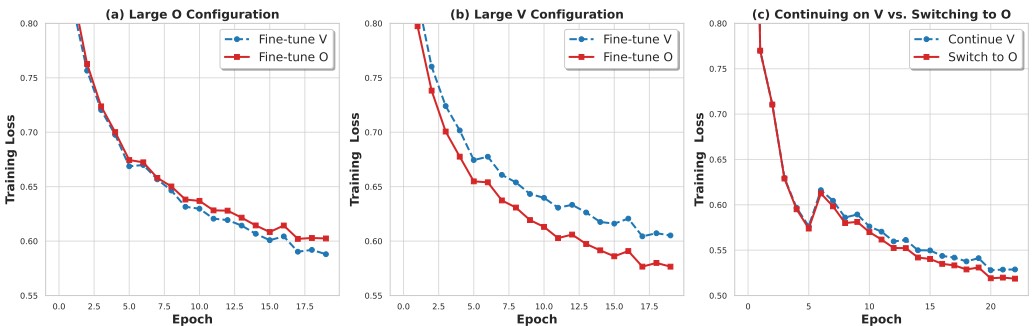

Figure 1: Effect of block norm on fine-tuning performance.

---

[2] https://github.com/google-research/vision_transformer

Furthermore, since the norms of trained blocks generally grow during training, it is possible to adopt a dynamic strategy that switches the target block partway through training. To test this, we first fine-tuned the $V$ block for five epochs, by which point its norm had become larger than that of $O$. As shown in subfigure (c), switching to the smaller $O$ block at this stage yields a modest but consistent improvement in performance. This experiment further validates that the relative matrix norm plays a key role in determining which block to tune.

## 4.2 SCALING UP TO LLAMA2-7B

Building on our findings from the vision datasets, we scale up our experiments to a larger backbone by evaluating the LLaMA2-7B model (Touvron et al., 2023) on a commonsense reasoning dataset originally studied in (Hu et al., 2023). Our experimental setup follows prior DoRA work (Liu et al., 2024), with one difference in label handling: whereas DoRA concatenates the instruction and label for next-word prediction, we instead prompt the model to predict the label directly.

Our primary interest is in the two-block setting: whether employing a more aggressive $OV$-tuning strategy outperforms the default $QV$ configuration used in conventional LoRA studies. To this end, we first fine-tune the model using only the $V$ block, then compare the effect of adding either $Q$ or $O$. The goal is to identify which additional block contributes more substantially to reductions in training and validation loss. Note that $Q$ and $O$ share the same dimensionality, so applying rank-1 LoRA introduces an equal number of trainable parameters. Nonetheless, results in Figure 2 demonstrate that augmenting the $V$ block with $O$ is more effective than adding $Q$, consistently

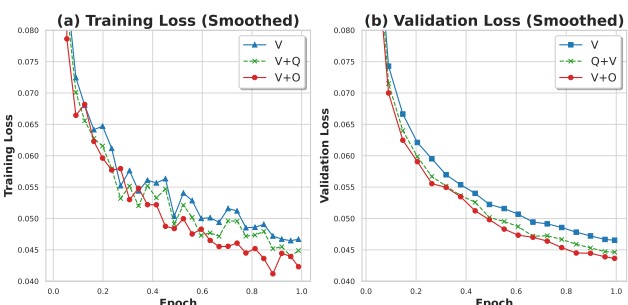

Figure 2: Comparison of two-block combinations on the LLaMA2-7B model. Fine-tuning the $V+O$ blocks achieves lower training and validation losses than $V+Q$. Results are shown for rank-1 LoRA, with training and validation curves smoothed using moving averages of 50 and 10 steps, respectively.

yielding lower training and validation losses. This improvement can be attributed in part to the generally smaller norm of the $O$ block (subfigure c) and, more importantly, to its operation outside the softmax function, which avoids potential constraints on tuning effectiveness. These findings are not unique to the rank-1 case, as confirmed by additional experiments in Appendix H.

Table 3: Comparison of LoRA block selection strategies on commonsense reasoning benchmarks with the LLaMA2-7B model. Scores are reported as accuracy (%).

| Algorithm | BoolQ | PIQA | SIQA | HellaSwag | WinoGrande | ARC-e | ARC-c | OBQA | Avg |
|---|---|---|---|---|---|---|---|---|---|
| LoRA (QK, Rank=1) | 68.41 | 80.03 | 77.64 | 81.75 | 78.54 | 84.97 | 67.41 | 72.00 | 76.34 |
| LoRA (QV, Rank=1) | 69.82 | 82.15 | 78.29 | **81.90** | 82.87 | 85.98 | 71.28 | 80.40 | 79.09 |
| LoRA (OV, Rank=1) | **70.70** | **83.03** | **80.04** | 81.82 | **83.74** | **86.36** | **71.59** | **80.80** | **79.76** |
| LoRA (QK, Rank=2) | 68.87 | 80.58 | 79.56 | 85.09 | 79.87 | 80.68 | 69.82 | 73.40 | 77.23 |
| LoRA (QV, Rank=2) | 70.70 | 82.65 | 79.20 | 87.64 | 82.32 | **86.21** | **71.87** | 80.40 | 80.12 |
| LoRA (OV, Rank=2) | **71.68** | **83.51** | **80.48** | **87.83** | **83.74** | 86.18 | 71.45 | **83.20** | **81.01** |
| LoRA (QK, Rank=4) | 71.12 | 82.10 | 79.31 | 84.25 | 80.43 | 81.56 | 70.74 | **83.60** | 79.14 |
| LoRA (QV, Rank=4) | 71.25 | 83.24 | **81.37** | 89.23 | 84.85 | 87.04 | **73.12** | 83.40 | 81.69 |
| LoRA (OV, Rank=4) | **72.29** | **84.44** | 80.14 | **90.25** | **85.24** | **87.75** | 72.10 | 84.40 | **82.08** |
| LoRA (QK, Rank=8) | 69.92 | 83.24 | 80.24 | 87.78 | 80.98 | 86.03 | 70.73 | 81.40 | 80.04 |
| LoRA (QV, Rank=8) | 71.83 | 83.90 | 81.42 | 89.88 | **85.35** | 87.33 | **73.12** | 85.00 | 82.23 |
| LoRA (OV, Rank=8) | **71.90** | **85.53** | **81.99** | **91.30** | 84.93 | **87.96** | 72.95 | **85.20** | **82.72** |

For further validations, Table 3 compares LoRA block selection across eight commonsense reasoning datasets. Across all rank settings, the $OV$ strategy consistently achieves the highest average scores, outperforming both the default $QV$ configuration and the opposite $QK$ setting. These results align with the training and validation curves in Figure 2, where tuning $O$ and $V$ leads to smaller losses. In contrast, the $QK$ configuration lags behind across nearly all tasks, echoing the patterns observed in our earlier ViT experiments. Taken together, the findings indicate that prioritizing the output and value blocks yields more effective representations for commonsense reasoning.

## 4.3 LLaMA3-8B Experiment

We further extend our analysis to the larger LLaMA3-8B model (Dubey et al., 2024), maintaining focus on the two-block tuning setting. Table 4 summarizes the evaluation results across all eight commonsense reasoning datasets, with corresponding training and validation loss curves provided in Appendix I. Overall, the observed patterns mirror those seen in the LLaMA2-7B experiments, with the $OV$-tuning strategy consistently outperforming the default $QV$ configuration. Notably, the advantage of $OV$ tuning is more pronounced in this larger model, particularly for lower-rank configurations. For instance, at rank 1, $OV$ surpasses $QV$ by 2.23%, while at rank 8, the performance gap narrows to 0.70%, echoing the trends observed in our ViT experiments in Table 1.

Table 4: Comparison of LoRA block selection strategies on commonsense reasoning benchmarks with the LLaMA3-8B model. Scores are reported as accuracy (%).

| Algorithm | BoolQ | PIQA | SIQA | HellaSwag | WinoGrande | ARC-e | ARC-c | OBQA | Avg |
|---|---|---|---|---|---|---|---|---|---|
| LoRA (QK, Rank=1) | 70.76 | 87.05 | 78.35 | 90.19 | 84.61 | 91.88 | 79.26 | 78.40 | 82.56 |
| LoRA (QV, Rank=1) | 66.36 | 86.62 | 79.89 | 92.65 | 86.12 | 92.00 | 78.41 | 85.40 | 83.43 |
| LoRA (OV, Rank=1) | **72.69** | **88.96** | **80.55** | **94.09** | **87.37** | **92.85** | **81.74** | **87.00** | **85.66** |
| LoRA (QK, Rank=2) | 71.71 | 88.74 | 78.92 | 91.32 | 85.24 | 91.62 | 78.41 | 81.40 | 83.42 |
| LoRA (QV, Rank=2) | 65.96 | 89.17 | 81.01 | 93.44 | 86.66 | 92.97 | 80.80 | 84.80 | 84.35 |
| LoRA (OV, Rank=2) | **73.55** | **90.04** | **81.68** | **94.69** | **88.48** | **93.22** | **82.08** | **87.60** | **86.42** |
| LoRA (QK, Rank=4) | 71.65 | 88.25 | 78.92 | 92.40 | 85.87 | 92.34 | 79.61 | 83.60 | 84.08 |
| LoRA (QV, Rank=4) | 73.79 | 89.23 | 82.04 | 94.38 | 88.48 | 93.31 | 81.14 | 87.40 | 86.22 |
| LoRA (OV, Rank=4) | **74.50** | **89.77** | **82.70** | **95.04** | **88.79** | 92.59 | 81.57 | **87.80** | **86.60** |
| LoRA (QK, Rank=8) | 72.39 | 88.96 | 79.89 | 93.33 | 86.42 | 92.30 | 80.55 | 86.20 | 85.01 |
| LoRA (QV, Rank=8) | 73.64 | 90.04 | 82.24 | 94.95 | **89.19** | **93.64** | 82.00 | 88.20 | 86.74 |
| LoRA (OV, Rank=8) | **74.86** | **90.42** | **83.52** | **95.92** | 89.11 | 93.27 | **82.25** | **90.20** | **87.44** |

## 4.4 Other PEFT Algorithms: AdaLoRA and DoRA

To demonstrate that our previous findings extend beyond the standard LoRA algorithm, we further evaluate two alternative PEFT methods: AdaLoRA and DoRA. Table 5 reports results for tuning either the value ($V$) or output ($O$) projection at rank-1 and rank-2, with higher-rank results provided in Appendix J. Overall, tuning $O$ generally outperforms $V$ across both algorithms and ranks, showing that this conclusion extends beyond vanilla LoRA to other PEFT frameworks. Specifically, while AdaLoRA introduces adaptive rank allocation and DoRA decouples magnitude and direction updates, both methods largely preserve the same relative ordering between $O$ and $V$. The only exception occurs with DoRA at rank-2, where tuning $V$ slightly surpasses $O$ by a minor margin of 0.04%.

Table 5: Performance of AdaLoRA and DoRA across different ranks and target blocks.

| Rank | Target | AdaLoRA | | | DoRA | | |
|---|---|---|---|---|---|---|---|
| | | CIFAR-100 | SVHN | Food-101 | CIFAR-100 | SVHN | Food-101 |
| 1 | V | $82.53_{\pm0.14}$ | $95.04_{\pm0.11}$ | $83.52_{\pm0.17}$ | $82.73_{\pm0.15}$ | $95.07_{\pm0.11}$ | $82.84_{\pm0.15}$ |
| | O | $\mathbf{83.09}_{\pm0.15}$ | $\mathbf{95.11}_{\pm0.16}$ | $\mathbf{83.95}_{\pm0.15}$ | $\mathbf{83.65}_{\pm0.10}$ | $\mathbf{95.14}_{\pm0.07}$ | $\mathbf{83.43}_{\pm0.18}$ |
| 2 | V | $83.05_{\pm0.11}$ | $95.30_{\pm0.11}$ | $84.65_{\pm0.12}$ | $83.80_{\pm0.13}$ | $\mathbf{95.80}_{\pm0.10}$ | $83.96_{\pm0.11}$ |
| | O | $\mathbf{83.95}_{\pm0.11}$ | $\mathbf{95.48}_{\pm0.10}$ | $\mathbf{85.07}_{\pm0.11}$ | $\mathbf{84.47}_{\pm0.09}$ | $95.76_{\pm0.10}$ | $\mathbf{84.27}_{\pm0.09}$ |

## 5 Conclusion

In this work, we revisit the fundamental question of block-level importance in parameter-efficient fine-tuning. Through a combination of theoretical analysis and extensive empirical evaluation, we highlight the critical role of the output block in class-token–dependent tasks, demonstrate that smaller-norm blocks can have amplified effects, and explain why tuning query and key blocks is often less impactful due to softmax damping. Furthermore, when tuning two blocks simultaneously, we show that prioritizing the output–value pair consistently outperforms the conventional query–value combination. These insights hold across multiple architectures, pretrained models, rank configurations, and downstream tasks, and generalize to other PEFT frameworks such as DoRA and AdaLoRA. Overall, our findings establish block selection as a fundamental design consideration in PEFT and offer practical, empirically grounded strategies for improving both the effectiveness and efficiency of model adaptation.

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

## 6 APPENDIX

### A PROOF FOR THEOREM 3

*Proof.* Since
$$S(W_Q, W_K) \;=\; \frac{1}{\sqrt{d}}\, X\, W_Q^\top W_K\, X^\top,$$

for perturbations $\Delta W_Q, \Delta W_K$ we have

$$D_{W_Q} S[\Delta W_Q] = \frac{1}{\sqrt{d}}\, X\, (\Delta W_Q)^\top W_K\, X^\top, \tag{A.1}$$

$$D_{W_K} S[\Delta W_K] = \frac{1}{\sqrt{d}}\, X\, W_Q^\top (\Delta W_K)\, X^\top. \tag{A.2}$$

Using $\|UVW\|_F \le \|U\|_2 \|V\|_F \|W\|_2$ and $\|X^\top\|_2 = \|X\|_2$,

$$\|D_{W_Q} S[\Delta W_Q]\|_F \le \frac{1}{\sqrt{d}}\, \|X\|_2\, \|(\Delta W_Q)^\top W_K\|_F\, \|X\|_2 \le \frac{1}{\sqrt{d}}\, \|X\|_2^2\, \|W_K\|_2\, \|\Delta W_Q\|_F, \tag{A.3}$$

$$\|D_{W_K} S[\Delta W_K]\|_F \le \frac{1}{\sqrt{d}}\, \|X\|_2^2\, \|W_Q\|_2\, \|\Delta W_K\|_F. \tag{A.4}$$

For a row $a_i = \mathrm{softmax}(s_i) \in \mathbb{R}^n$, its Jacobian is $J(a_i) = \mathrm{diag}(a_i) - a_i a_i^\top$. We then have:

$$\|J(a_i)\|_2 \;\le\; 1 - \|a_i\|_2^2 \;=\; (1 + \|a_i\|_2)(1 - \|a_i\|_2) \;\le\; 2(1 - a_{i,\max}).$$

We then have:

$$a_{i,\max} = \frac{1}{1 + \sum_{j \ne \arg\max} \exp(-(s_{i,\max} - s_{i,j}))} \ge \frac{1}{1 + (n-1)e^{-\gamma_i}},$$

hence $1 - a_{i,\max} \le \min\{(n-1)e^{-\gamma_i}, 1\}$ and

$$\|J(a_i)\|_2 \;\le\; 2\min\{(n-1)\, e^{-\gamma_{\min}}, 1\}. \tag{A.5}$$

Because the row-softmax acts independently on rows, its derivative $D_{\mathrm{softmax}(S)}[\cdot]$ is block-diagonal with blocks $J(a_i)$; thus

$$\|D_{\mathrm{softmax}(S)}[\Delta S]\|_F \;\le\; 2\min\{(n-1)\, e^{-\gamma_{\min}}, 1\}\, \|\Delta S\|_F. \tag{A.6}$$

Regard $F$ as the composition

$$(S \mapsto A = \mathrm{softmax}(S)) \quad \text{then} \quad (A \mapsto F = AXW_V W_O).$$

The derivative of the second map at $(W_V, W_O, X)$ is

$$D_A F[\Delta A] \;=\; \Delta A X W_V W_O,$$

hence

$$\|D_A F[\Delta A]\|_F \;\leq\; \|W_O\|_2 \|W_V\|_2 \|X\|_2 \|\Delta A\|_F. \tag{A.7}$$

Combining equation A.6–equation A.7 with equation A.3–equation A.4 and the chain rule yields

$$\frac{\|D_{W_Q} F[\Delta W_Q]\|_F}{\|\Delta W_Q\|_F} \leq \frac{2(n-1)e^{-\gamma_{\min}}}{\sqrt{d}} \|W_O\|_2 \|W_V\|_2 \|W_K\|_2 \|X\|_2^3,$$

$$\frac{\|D_{W_K} F[\Delta W_K]\|_F}{\|\Delta W_K\|_F} \leq \frac{2(n-1)e^{-\gamma_{\min}}}{\sqrt{d}} \|W_O\|_2 \|W_V\|_2 \|W_Q\|_2 \|X\|_2^3.$$

For $W_V$ and $W_O$, direct differentiation gives

$$D_{W_V} F[\Delta W_V] = AX\Delta W_V W_O, \qquad D_{W_O} F[\Delta W_O] = AXW_V \Delta W_O,$$

hence, using $\|UVW\|_F \leq \|U\|_2 \|V\|_F \|W\|_2$,

$$\frac{\|D_{W_V} F[\Delta W_V]\|_F}{\|\Delta W_V\|_F} \;\leq\; \|W_O\|_2 \|A\|_2 \|X\|_2, \qquad \frac{\|D_{W_O} F[\Delta W_O]\|_F}{\|\Delta W_O\|_F} \;\leq\; \|A\|_2 \|W_V\|_2 \|X\|_2.$$

$\square$

## B  PROOF FOR THEOREM 4

*Proof.* According to the standard estimation for spherical caps, we have that

$$\Pr(|\langle u, v\rangle| \geq t) \leq 2e^{-(d-1)t^2/2}, \tag{B.1}$$

for $u, v$ drawn independently and uniformly from the unit sphere $\mathbb{S}^{d-1}$. Therefore, we have

$$\Pr\left(\max_{i \neq j} |\langle x_i, x_j\rangle| > t_\delta\right) \leq 2e^{-(d-1)t_\delta^2/2} \cdot n(n-1)/2 \;<\; \delta, \tag{B.2}$$

On this event, for all $i \neq j$ we have $|S_{ij}| \leq \frac{1}{\sqrt{d}}\|M\|_2\, t_\delta \leq \frac{\beta c^2}{\sqrt{d}} t_\delta$, while $S_{ii} = \frac{1}{\sqrt{d}} x_i^\top M x_i \geq \frac{1}{\sqrt{d}}\lambda_{\min}(M_{\text{sym}}) \geq \frac{\alpha c^2}{\sqrt{d}}$. Thus every row margin satisfies

$$\gamma_{\min} \;\geq\; \frac{(\alpha - \beta t_\delta)c^2}{\sqrt{d}} \;=\; a\,c^2. \tag{B.3}$$

Apply the Gershgorin's theorem on $XX^\top$ with unit diagonal and off-diagonals $\leq t_\delta$, we have that

$$\|X\|_2^2 = \|XX^\top\|_2 \leq 1 + (n-1)t_\delta = \chi_\delta^2 \tag{B.4}$$

From the row-softmax Jacobian bound (per-row $\|J\|_2 \leq 2(n-1)e^{-\gamma}$) and the chain rule (as proved earlier), for equal-norm perturbations and scale comparability we have

$$\frac{\|D_{W_Q} F\|_F}{\|D_{W_V} F\|_F} \;\leq\; \frac{\min\{2(n-1)e^{-\gamma_{\min}}, 1\}}{\sqrt{d}} \frac{\tau^2 c^2 \|X\|_2^2}{\|A\|_2} \;\leq\; \min\{2(n-1)e^{-ac^2}, 1\} \cdot \frac{\tau^2 c^2 \chi_\delta^2}{\sqrt{d}},$$

and the analogous inequalities for the other three ratios (using $\|A\|_2 \geq 1$). When $ac^2 > \log(2(n-1))$, the ratio is bounded by

$$R(c) \;=\; 2(n-1)\,e^{-ac^2} \cdot \frac{\tau^2 c^2 \chi_\delta^2}{\sqrt{d}} \;=\; \frac{2(n-1)\tau^2 \chi_\delta^2}{\sqrt{d}} \cdot \underbrace{\left(c^2 e^{-ac^2}\right)}_{=:g(c)}. \tag{B.5}$$

Let $y := ac^2$. Then, $g(c) = \frac{y}{a}e^{-y}$ and the condition $R(c) \leq \eta$ is

$$y\,e^{-y} \;\leq\; \frac{\eta\,a\sqrt{d}}{2(n-1)\tau^2 \chi_\delta^2} \;=:\; K. \tag{B.6}$$

For $K \in (0, 1/e)$, this inequality is equivalent to $y \geq -W_{-1}(-K)$ (the $-1$ branch). Taking $y \geq \max\{\log(2(n-1)), -W_{-1}(-K)\}$ and then $c^2 \geq y/a$ gives the desired domination inequalities.

$\square$

## C  PROOF FOR THEOREM 5

*Proof.* Consider the $i$-th token output row $F_i^\top = b_i^\top W_V W_O$.

**(1) Perturbing $W_V$:** Let $\Delta W_V$ be an arbitrary perturbation. The first-order change in $F_i^\top$ is

$$\Delta F_i^\top = b_i^\top \Delta W_V W_O.$$

Since $\Delta W_V$ is multiplied on the right by $W_O$, the resulting vector $\Delta F_i^\top$ is always a linear combination of the rows of $W_O$. Therefore,

$$\Delta F_i^\top \in \mathrm{row}(W_O) \subseteq \mathbb{R}^{1\times d}.$$

This shows that perturbations to $W_V$ cannot move $F_i^\top$ outside the row space of $W_O$.

**(2) Perturbing $W_O$:** Let $\Delta y \in \mathbb{R}^{1\times d}$ be any desired target change. If $b_i^\top W_V \neq 0$, the Moore-Penrose pseudoinverse $(b_i^\top W_V)^+$ exists and satisfies

$$b_i^\top W_V (b_i^\top W_V)^+ \Delta y = \Delta y.$$

Define

$$\Delta W_O := (b_i^\top W_V)^+ \Delta y.$$

Then

$$\Delta F_i^\top = b_i^\top W_V \Delta W_O = b_i^\top W_V (b_i^\top W_V)^+ \Delta y = \Delta y,$$

showing that any desired output change $\Delta y$ can be achieved by a suitable choice of $\Delta W_O$ as long as $b_i^\top W_V \neq 0$.

**Conclusion:** For a specific token $i$, perturbing $W_V$ can only produce changes within $\mathrm{row}(W_O)$, whereas perturbing $W_O$ can realize arbitrary directions in $\mathbb{R}^{1\times d}$, provided $b_i^\top W_V \neq 0$.  $\square$

## D  EXPERIMENT CONFIGURATIONS

We conduct experiments on both vision and language tasks to examine the impact of block selection in fine-tuning. For vision tasks, we follow the design of AdaptFormer, focusing on classification across multiple datasets with a ViT-B16 backbone. Note we adopt a different pretrained ViT model, chosen for its stronger overall performance. All ViT experiments use a fixed learning rate of $1\times 10^{-3}$ with an exponential decay factor of 0.9 per epoch, and models are fine-tuned for 20 epochs in total. For language model experiments, we set the learning rate to $1\times 10^{-4}$ for LLaMA3-8B and $2\times 10^{-4}$ for LLaMA2-7B, with a linear decay to zero following the setup used in prior DoRA studies. Across all experiments, we employ the AdamW optimizer (Loshchilov & Hutter, 2017) with a weight decay of 0.1. To investigate the role of block selection, we evaluate both single-block and two-block configurations. Table 6 summarizes our configurations for all tasks.

Table 6: Experimental Setup for Vision and Language Tasks.

| Task | PEFT | Model | LR | Schedule / Epochs | Optimizer | Weight Decay |
|------|------|-------|-----|------------------|-----------|-------------|
| Vision | LoRA | ViT-B16 (MAE) | $1\times 10^{-3}$ | Exponential decay 0.9 / 50 epochs | AdamW | 0.1 |
| Vision | LoRA | ViT-B16 (ImageNet) | $1\times 10^{-3}$ | Exponential decay 0.9 / 20 epochs | AdamW | 0.1 |
| Vision | AdaLoRA | ViT-B16 (MAE) | $1\times 10^{-3}$ | Exponential decay 0.9 / 50 epochs | AdamW | 0.1 |
| Vision | DoRA | ViT-B16 (MAE) | $1\times 10^{-3}$ | Exponential decay 0.9 / 50 epochs | AdamW | 0.1 |
| Language | LoRA | LLaMA2-7B | $2\times 10^{-4}$ | Linear decay to 0 / - | AdamW | 0.1 |
| Language | LoRA | LLaMA3-8B | $1\times 10^{-4}$ | Linear decay to 0 / - | AdamW | 0.1 |

**Block Selection:** Single-block and two-block configurations evaluated across all experiments.

# E   VIT ADDITIONAL EXPERIMENTS - SINGLE-BLOCK TUNING

To further validate on single-block tuning, we conduct additional experiments on the ViT-B16 backbone across multiple image classification datasets. In particular, we present three complementary subfigures: (a) the training loss curves for each individual block, (b) the corresponding test accuracy, and (c) the Frobenius norm of each block's weight matrix. These results allow us to analyze not only how each block affects optimization dynamics and final performance, but also how the intrinsic scale of the block (as captured by the Frobenius norm) relates to its influence on downstream tasks.

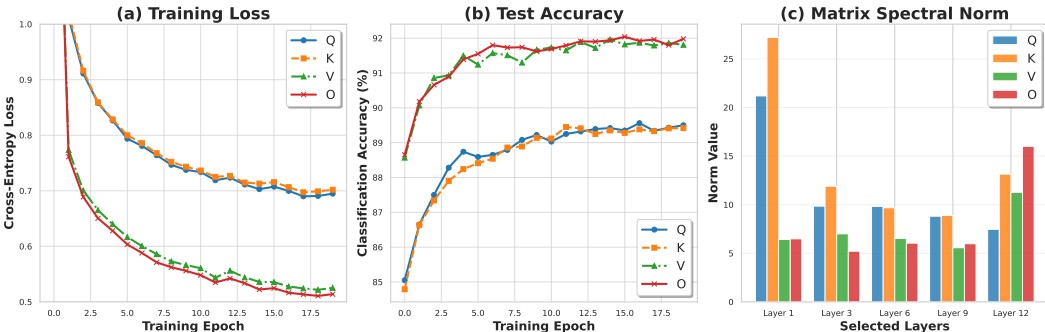

Figure 3: Effect of block selection strategies on rank-4 PEFT performance. The first two sub-figures show training and test dynamics, while the last figure compares the spectral norm of modules on each layer.

Figure 3 presents the training and testing dynamics, along with a comparison of the matrix norms across different modules and layers. The results show that tuning the $O$ and $V$ blocks consistently yields superior performance, in line with our earlier analysis. Since the Frobenius norms of $O$ and $V$ are often comparable in the pretrained model, their performance differences are generally small. Nevertheless, $O$ tends to outperform $V$ slightly. This observation aligns with our previous finding that when final performance depends heavily on the class token, the $O$ block plays a more critical role. Importantly, in terms of test accuracy, the performance gap can reach up to 2.5% when comparing $O$ against $Q$ or $K$, highlighting the significance of carefully selecting which blocks to tune in LoRA.

# F   VIT ADDITIONAL EXPERIMENTS - TWO-BLOCK TUNING

Building on the single-block tuning experiments presented in the main text, we further investigate two-block configurations in vision transformers. These experiments use a supervised-pretrained ViT-B16 model on ImageNet-21K and focus on evaluating how combinations of blocks affect training dynamics and downstream performance. The results provide complementary insights to the single-block studies and help validate the generality of the block-selection principles across multi-block configurations.

Here we focus on the two-block setting, which involves jointly tuning pairs of attention components within each layer. Note in the standard LoRA configuration, the $Q$ and $V$ blocks are tuned. Building on our previous analysis, we propose an alternative strategy that tunes $V$ and $O$, and additionally examine the opposite configuration of tuning $Q$ and $K$. Comparing these strategies allows us to systematically evaluate the impact of block selection on fine-tuning performance.

Table 7 presents the performance of different block selection strategies for LoRA fine-tuning across multiple datasets and rank configurations. Across all datasets and ranks, the proposed $OV$ strategy consistently outperforms both the default $QV$ configuration and the contrast $QK$ setting, achieving the highest accuracy in every case. While the default $QV$ configuration remains competitive, the contrast $QK$ strategy generally yields the lowest performance, highlighting that not all block choices contribute equally to effective fine-tuning. These results confirm that careful selection of attention blocks, specifically prioritizing the $O$ and $V$ blocks, can consistently improve downstream task performance and validate the effectiveness of our proposed strategy.

Table 7: Comparison of block selection strategies for LoRA fine-tuning across datasets. The "Default" column indicates the standard LoRA configuration by choosing $Q$ and $V$ blocks, "Proposed" highlights our $OV$ strategy, and "Contrast" corresponds to the $QK$ configuration used for comparison. Performance is reported as mean $\pm$ standard deviation.

| Configuration | Strategy | # Parameters | CIFAR-100 | SVHN | Food-101 |
|---|---|---|---|---|---|
| LoRA (QK, Rank-1) | Contrast | 0.11 M | $89.25_{\pm 0.16}$ | $93.12_{\pm 0.17}$ | $87.64_{\pm 0.21}$ |
| LoRA (QV, Rank-1) | Default | 0.11 M | $91.40_{\pm 0.06}$ | $95.32_{\pm 0.10}$ | $89.80_{\pm 0.19}$ |
| LoRA (OV, Rank-1) | Proposed | 0.11 M | $\mathbf{91.62}_{\pm 0.07}$ | $\mathbf{95.79}_{\pm 0.11}$ | $\mathbf{89.92}_{\pm 0.13}$ |
| LoRA (QK, Rank-2) | Contrast | 0.15 M | $89.75_{\pm 0.14}$ | $94.32_{\pm 0.17}$ | $88.30_{\pm 0.17}$ |
| LoRA (QV, Rank-2) | Default | 0.15 M | $91.85_{\pm 0.13}$ | $96.16_{\pm 0.11}$ | $90.10_{\pm 0.09}$ |
| LoRA (OV, Rank-2) | Proposed | 0.15 M | $\mathbf{92.11}_{\pm 0.10}$ | $\mathbf{96.21}_{\pm 0.08}$ | $\mathbf{90.26}_{\pm 0.06}$ |
| LoRA (QK, Rank-4) | Contrast | 0.22 M | $90.20_{\pm 0.14}$ | $95.32_{\pm 0.14}$ | $88.81_{\pm 0.20}$ |
| LoRA (QV, Rank-4) | Default | 0.22 M | $92.09_{\pm 0.06}$ | $96.48_{\pm 0.10}$ | $90.31_{\pm 0.11}$ |
| LoRA (OV, Rank-4) | Proposed | 0.22 M | $\mathbf{92.13}_{\pm 0.04}$ | $\mathbf{96.72}_{\pm 0.07}$ | $\mathbf{90.61}_{\pm 0.07}$ |
| LoRA (QK, Rank-8) | Contrast | 0.37 M | $90.46_{\pm 0.10}$ | $95.77_{\pm 0.10}$ | $89.22_{\pm 0.09}$ |
| LoRA (QV, Rank-8) | Default | 0.37 M | $92.03_{\pm 0.05}$ | $96.96_{\pm 0.06}$ | $90.65_{\pm 0.06}$ |
| LoRA (OV, Rank-8) | Proposed | 0.37 M | $\mathbf{92.23}_{\pm 0.05}$ | $\mathbf{97.15}_{\pm 0.09}$ | $\mathbf{90.81}_{\pm 0.04}$ |

Introducing a second block in the tuning process generally yields a modest improvement in performance, though the magnitude of this gain depends on the rank configuration. For instance, when tuning rank-1 LoRA modules on ViT-B16, selecting the output ($O$) block alone already achieves high accuracy on CIFAR-100, SVHN, and Food-101, but combining the output and value ($OV$) blocks leads to a slight increase in accuracy, typically on the order of 0.2–0.5%. As the rank increases to 4 or 8, this additional gain becomes even smaller, with improvements often below 0.2%, indicating diminishing returns from tuning multiple blocks simultaneously. These patterns suggest that, for classification tasks with ViT architectures, single-block tuning, particularly of the output block, is generally sufficient to capture most of the performance benefits. Two-block tuning can still provide incremental gains, but the added complexity and parameter overhead may not justify the marginal improvement in accuracy, especially in resource-constrained scenarios. Overall, these results reinforce the principle that carefully selecting the most impactful block is more important than simply increasing the number of blocks tuned.

## G LLAMA2-7B ADDITIONAL EXPERIMENT - SINGLE BLOCK

In addition to our ViT studies, we conduct further experiments on the LLaMA2-7B language model to investigate the effects of single-block tuning in a large-scale language setting. Specifically, we evaluate the training dynamics and relative contributions of individual attention blocks by tracking three key metrics: the training loss, validation loss, and the Frobenius norm of the pretrained attention matrices for each block. These measurements allow us to compare the effectiveness of tuning query, key, value, and output projections, providing insight into which blocks are most influential for adaptation in language tasks. The results also serve to complement our theoretical analysis and vision experiments.

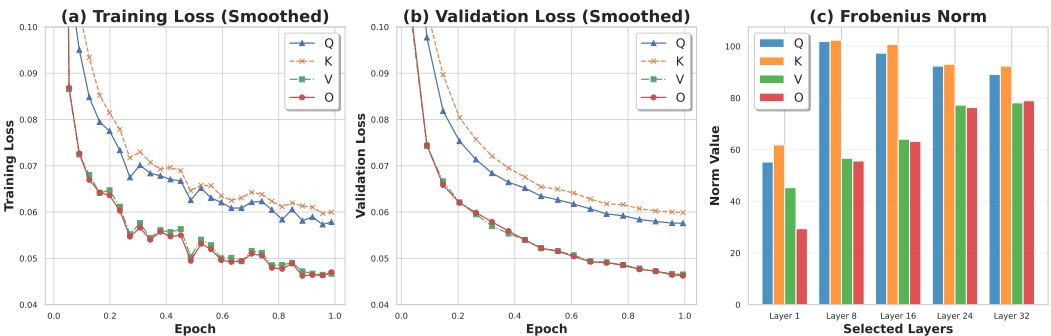

Figure 4: Comparison of single-block tuning with LlaMA2-7B model. Rank is 1 for all blocks.

The results in Figure 4 reveal several noteworthy patterns. First, the training curves for the value ($V$) and output ($O$) blocks are closely aligned, reflecting their similar Frobenius norms and comparable

contributions to adaptation. Unlike the ViT experiments, which are highly dependent on the class token, the LLaMA2-7B language model does not rely on a single token representation, yet the similarity between $V$ and $O$ remains evident. Second, the query ($Q$) and key ($K$) blocks exhibit slower training progress and higher losses compared to $V$ and $O$, consistent with the dampening effect of the softmax operation. Notably, the $K$ block, which has a relatively larger norm, shows slightly slower convergence than $Q$, further confirming that block norms influence training dynamics. Overall, these observations mirror the trends seen in the ViT experiments, suggesting that the relative importance of blocks, favoring $O$ and $V$ over $Q$ and $K$, is a general phenomenon across both vision and language transformer models.

## H    LLaMA2-7B ADDITIONAL EXPERIMENT - TWO BLOCKS

To complement the main text, we present additional experiments on two-block tuning for LLaMA2-7B using rank-2 adaptations. We first illustrate the experimental setup for two-block tuning at rank-2. The goal is to compare two configurations: fine-tuning the value ($V$) block together with the query ($Q$) block versus fine-tuning $V$ together with the output ($O$) block. This setup allows us to directly examine the impact of including the output block on training dynamics and final performance.

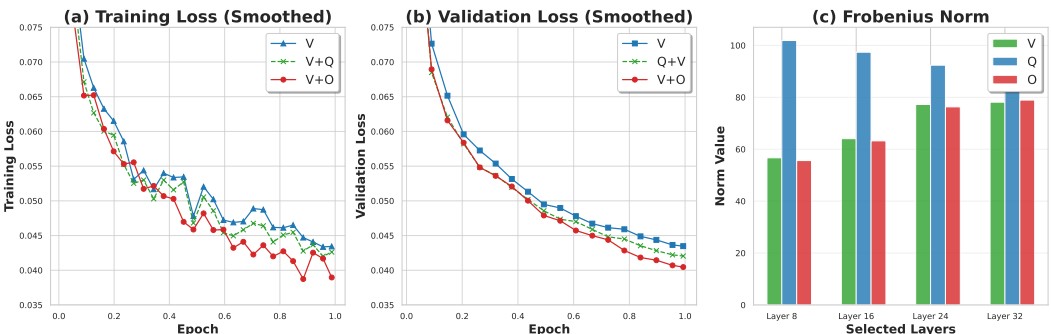

Figure 5: Comparison of two-block tuning with LlaMA2-7B model. Rank is 2 for all blocks.

Figure 5 presents the results of these two-block configurations. Subfigure (a) shows the training loss over epochs, subfigure (b) shows the validation loss, and subfigure (c) reports the Frobenius norms of the selected blocks. Consistent with the rank-1 experiments, including the output block ($V$+$O$) leads to faster loss reduction and slightly lower final losses compared to the $V$+$Q$ combination. This indicates that the benefit of prioritizing the output block extends to higher-rank adaptations, reinforcing the generality of our block-selection principle.

## I    LLaMA3-8B ADDITIONAL EXPERIMENT - TWO BLOCKS

We also conduct two-block tuning experiments on LLaMA3-8B to verify whether the trends observed in LLaMA2-7B generalize to a larger model. Similar to the previous experiments, we compare the $V$+$O$ and $V$+$Q$ configurations. The results show a consistent pattern: including the output block ($V$+$O$) leads to faster reduction in both training and validation losses compared to the $V$+$Q$ combination. The Frobenius norms of the selected blocks again reveal that $V$ and $O$ have comparable magnitudes, while the query block exhibits a smaller impact on loss decrease. Overall, these findings reinforce the generality of our block-selection principle across model sizes, confirming that prioritizing the output block is a robust strategy for effective two-block tuning in LLaMA-family models.

We do not include single-block tuning experiments for LLaMA3-8B in this study. This is because the model employs a group-query attention mechanism (Ainslie et al., 2023; Shazeer, 2019), which results in the key ($K$) and value ($V$) projections having different dimensions from the query ($Q$) and output ($O$) projections. Consequently, tuning only a single block would lead to inconsistent parameter counts and complicate direct comparisons between blocks. By focusing on two-block configurations with the same number of trainable parameters, we ensure a fair evaluation while preserving the consistency of the low-rank adaptation across attention projections.

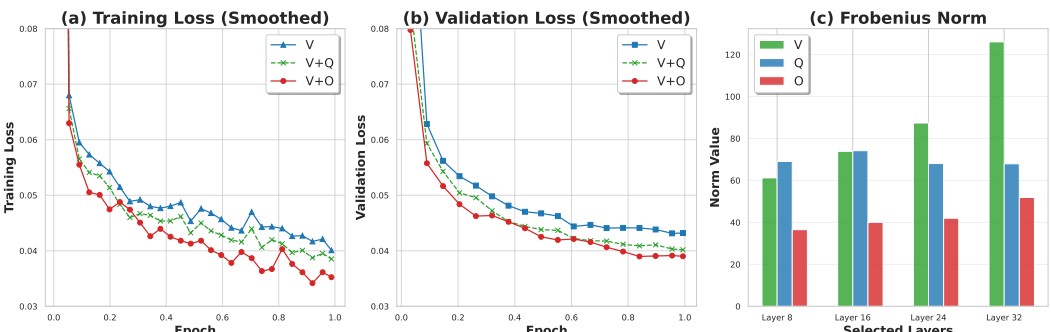

Figure 6: Comparison of two-block tuning with LlaMA3-8B model. Rank is 1 for all blocks.

## J  ADDITIONAL RESULTS ON OTHER PEFT ALGORITHMS

To complement the main results, Table 8 presents the performance of AdaLoRA and DoRA at higher ranks (4 and 8), comparing tuning the value ($V$) versus output ($O$) blocks. Consistent with the trends observed at lower ranks, tuning the output block generally yields better performance across datasets and algorithms. For example, for rank-8, AdaLoRA achieves $85.23\%$ on CIFAR-100 when tuning $O$, compared to $84.74\%$ when tuning $V$. Similarly, DoRA demonstrates a consistent advantage of $O$-tuning across most tasks, with gains up to $0.15\%$ in Food-101. These results indicate that the principle of prioritizing the output block extends beyond LoRA to other PEFT frameworks, confirming the broader applicability of our block-selection insights.

However, there is one exception: for DoRA at rank-4 on SVHN, tuning the value block slightly outperforms the output block by a small margin ($96.46\%$ vs. $96.41\%$), demonstrating that while output-block prioritization is generally effective, specific combinations of algorithm, dataset, and rank can occasionally favor $V$-tuning. Overall, across both AdaLoRA and DoRA, and across all tested ranks, the empirical evidence strongly supports the relative importance of the output block, with the value block serving as a useful complement when multiple blocks can be tuned simultaneously.

Table 8: Performance of AdaLoRA and DoRA at higher ranks.

| Rank | Target | AdaLoRA | | | DoRA | | |
|---|---|---|---|---|---|---|---|
| | | CIFAR-100 | SVHN | Food-101 | CIFAR-100 | SVHN | Food-101 |
| 4 | V | $84.24_{\pm0.10}$ | $96.00_{\pm0.07}$ | $85.60_{\pm0.11}$ | $84.65_{\pm0.09}$ | $\mathbf{96.46}_{\pm0.07}$ | $85.31_{\pm0.10}$ |
| | O | $\mathbf{84.31}_{\pm0.09}$ | $\mathbf{96.15}_{\pm0.07}$ | $\mathbf{85.91}_{\pm0.10}$ | $\mathbf{84.71}_{\pm0.04}$ | $96.41_{\pm0.07}$ | $\mathbf{85.37}_{\pm0.06}$ |
| 8 | V | $84.74_{\pm0.09}$ | $96.37_{\pm0.06}$ | $86.41_{\pm0.10}$ | $85.44_{\pm0.04}$ | $96.81_{\pm0.04}$ | $86.58_{\pm0.05}$ |
| | O | $\mathbf{85.23}_{\pm0.07}$ | $\mathbf{96.54}_{\pm0.08}$ | $\mathbf{86.67}_{\pm0.07}$ | $\mathbf{85.57}_{\pm0.05}$ | $\mathbf{96.89}_{\pm0.05}$ | $\mathbf{86.61}_{\pm0.05}$ |

## K  USE OF LLMS

In preparing this manuscript, we utilized a large language model (ChatGPT by OpenAI) to assist in refining and polishing the text. Specifically, the LLM was employed to:

- Enhance clarity, coherence, and conciseness of the draft.
- Rephrase sentences to improve grammatical correctness and overall readability.
- Ensure consistent terminology and smooth transitions throughout the manuscript.

All LLM-generated suggestions were reviewed, edited, and verified by the authors for technical accuracy, logical consistency, and fidelity to the research content. No LLM outputs were used without human oversight. Importantly, the LLM was not used for data generation, model training or experiment design.

