# OpenReview forum: "LoRA in the Right Place: Which Block to Tune in Parameter-Efficient Fine-Tuning?"
_ICLR.cc/2026/Conference — ICLR 2026 Conference Withdrawn Submission_

### Official Review · Reviewer_r3m5 · 2025-10-21

**Soundness:** 2
**Presentation:** 2
**Contribution:** 1
**Rating:** 4
**Confidence:** 4

**Summary:**

In this work, the authors propose a criterion to choose which blocks of a transformer to give priority to for LoRA fine-tuning. In particular, the authors propose a sensitivity-based approach to identify the most important adapters in a self-attention block.

**Strengths:**

The work is overall well written and the problem that the authors challenge is definitely important from an application point of view.

**Weaknesses:**

1. While sensitivity analysis and error propagation through a transformer block might give some insights, I am not sure it gives the full picture on which layers it is better to spend parameters. In particular, to properly allocate parameters, one should also look at the sensitivity of the overall loss with respect to the full block: in fact, the toy example in Proposition 1 exactly reflects this, that the gradient of the objective function is bigger in one direction, and not the directional derivative of a layer itself. For this reason, the analysis feels partially completed, as in an overall architecture, each block's importance should probably be reweighted with $\frac{\partial}{\partial Z_l} \mathcal L$.

2. I believe in general the problem is not even well posed from a statistical point of view, i.e., I can construct a data model for which it is optimal to adapt $W_o$ and another one for which it is optimal to adapt $W_v$, for example. For this reason,  the problem of where to allocate the additional fine-tuning parameters depends on the data distribution, and it is not a pure feature of the model itself. The conclusion of the work cannot therefore be considered per-sé universal, but the authors don't propose an end-to-end procedure to choose which layers and which parameters of the layer to fine-tune.

4. Concerning Theorem 5, of course, if the model of the kind $XW_v W_o$, then the image is contained in the row space of $W_o$, but this is a simple consequence of linearity. Moreover, the results are presented for a single token, while the interesting part is what can be jointly represented. These spaces can be described explicitly, as it is simply the set of matrices $Y$ such that $range(Y) \subseteq range(X), rank(Y) \leq r $, where $r$ is the smallest dimension of the matrices (the bottleneck of the representation) [1,2].

3. Concerning numerical results, given the discussion above, I believe they should be contextualized more, as it really depends on the specific benchmark at hand.

[1] K. Kohn, "The Geometry of the Neuromanifold", SIAM news, Volume 57 Issue 06 July/August 2024.
[2] K. Kubjas et al., "Geometry of Polynomial neural networks, ArXiv 2024.

**Questions:**

I would appreciate hearing the author's comment on the weaknesses, as I believe the current work could be significantly improved following those suggestions.

---

### Official Review · Reviewer_m1qY · 2025-10-31

**Soundness:** 2
**Presentation:** 2
**Contribution:** 2
**Rating:** 2
**Confidence:** 3

**Summary:**

The paper investigates which components in an attention block should be adapted in PEFT frameworks such as LoRA, AdaLoRA, and DoRA. The authors present a theoretical analysis based on scalar and matrix perturbation formulations, extending to full transformer attention. They show that smaller-norm matrices yield amplified perturbation effects, while the softmax operation suppresses updates to the query and key blocks. From this, they derive a sensitivity hierarchy $W_O \ge W_V \gg W_Q \approx W_K$, establishing that the output and value projections contribute most to effective adaptation. Empirical results on ViT-B/16, LLaMA2-7B, and LLaMA3-8B validate the theory, with the output–value $(O+V)$ configuration consistently outperforming the conventional query–value $(Q+V)$ setup across ranks and datasets. The study aims to replace heuristic choices with theoretically grounded block selection principles for more efficient and effective PEFT.

**Strengths:**

The paper provides a clear theoretical foundation for block selection in PEFT and links sensitivity analysis to practical fine-tuning design. The proposed sensitivity hierarchy offers a simple yet generalizable rule applicable across PEFT variants. Overall, the work’s strength lies in its interpretable analysis and in demonstrating that a long-standing heuristic decision in LoRA can be optimized through formal reasoning.

**Weaknesses:**

1. **Weak theoretical rigor**

Both analyses rely solely on norm inequalities, e.g.

$$
\frac{\|\Delta W_1 W_2\|_F}{\|\Delta W_1\|_F} \le \|W_2\|_2, \quad
\frac{\|W_1 \Delta W_2\|_F}{\|\Delta W_2\|_F} \le \|W_1\|_2,
$$

and later

$$
\|D_{W_Q}F[\Delta W_Q]\|_F \le c_Q \|W_O\|_2\|W_V\|_2\|W_K\|_2.
$$

These expressions treat perturbation effects purely through spectral magnitudes, ignoring the alignment between singular subspaces of matrices and the gradient directions. As a result, the derivations represent loose worst-case upper bounds and fail to capture realistic sensitivity governed by cross-matrix orientation or task-specific gradient structure.

In Proposition 2, the setup

$$
W = W_1 W_2, \quad \text{with perturbations } \Delta W_1, \Delta W_2
$$

abstracts the idea of one layer’s weight being multiplied by another, similar to sequential linear maps. However, this is a highly simplified linear composition—it ignores softmax normalization, residual connections, and multi-head attention coupling. In real attention modules, gradients flow through non-linear and interdependent pathways

$$
F(X) = \text{softmax}(XW_Q^\top W_K X^\top / \sqrt{d})XW_VW_O,
$$

so the independent perturbation assumption in \(W = W_1W_2\) is unrealistic. The scalar or two-matrix setting provides intuition but does not correspond to any direct operation in actual transformer attention.

2. **Restricted evaluation scope.**
The experiments primarily rely on LoRA-based fine-tuning, with AdaLoRA and DoRA appearing only in a small supplementary comparison (Table 5) where the performance gain seems indefinite. The main results (Tables 1–4) evaluate only LoRA configurations across ViT-B/16 and LLaMA models.

3. **Marginal performance improvement**
Although consistent across settings, the reported gains are minimal under the practical rank-8 configuration—the most common LoRA setup. For example, Table 4 shows OV tuning outperforms QV by only about 0.5 % on LLaMA3-8B, and similar marginal gaps appear in Table 1 for ViT-B/16. No statistical test is provided to verify whether such small improvements are significant.

4. **Lack of empirical analysis or visualization.**
Although the paper emphasizes theoretical sensitivity analysis (Theorem 3–5), it provides no empirical evidence showing how this sensitivity manifests in practice. Figures such as 1–3 only report training loss and accuracy, without examining gradient magnitudes, attention activations, or Jacobian norms per block. Consequently, the link between the proposed theoretical hierarchy and real model behavior remains unvalidated.

**Questions:**

1. How would the theoretical framework change if the analysis incorporated subspace alignment or gradient-directional sensitivity rather than relying solely on spectral norm bounds?
2. Can the authors extend the theoretical derivation beyond the simplified linear composition $W = W_1W_2$ to a more realistic multi-head attention formulation that includes softmax coupling and residual connections?
3. Given that experiments primarily rely on LoRA under rank-8 settings with small performance gaps (~0.5%), can the authors provide statistical validation or variance analysis to confirm that the reported improvements are significant?
4. Could the authors include empirical analysis, such as per-component gradient norms or activation Jacobians in attention block, to demonstrate that the theoretical “sensitivity hierarchy” is observable during actual fine-tuning?

---

### Official Review · Reviewer_P67i · 2025-10-31

**Soundness:** 4
**Presentation:** 4
**Contribution:** 3
**Rating:** 6
**Confidence:** 4

**Summary:**

PEFT aims to solve the issue of training models with a smaller number of parameters than that of the whole model while retaining as much performance as possible. By doing this to every linear layer, certain blocks may be important to fine-tune, while others can be completely omitted. This work addresses the latter, developing theory to underpin the choice of blocks to fine-tune. This work investigates the difference between updating the different parameters of attention and their relative effects on the outputs of the model. Experimentally, it is shown that for ViTs, this hierarchy of effect directly translates to relatively performance when only that parameter is updated. These results translate to other language models along with other fine-tuning methods.

**Strengths:**

- While the difference between the KQ and OV circuits are well established and some anecdotal evidence around which to tune and what effect that will have, this work provides clear theoretical backing for how the model will be affected by similar changes to each of these matrices. This includes the theorem detailing how the OV circuit is asymmetric, with updates to V being throttled by the properties of O
- The motivation for the more lengthy theorems is quite clear, building from simple toy models in one dimension all the way to attention mechanisms
- Theorem 4 is shown to apply to concrete values in a common foundation model, rather than being left to be some abstract claim that may be vacuous
- Experimental results show cleanly the expected comparison in performance based on the described theory. These include validations that relative scale of V and O determine the difference in fine-tuning these block
- With these results

**Weaknesses:**

- The theory develops around a single block improvements in a single layer, but the experiments are performed by fine-tuning every layer of a specific kind in the whole model. Some single block/single layer experiments on certain layers would connect the theory and experiments even more strongly
- Anecdotally, updating parameters in the MLPs of the model often yields larger changes in performance. This is mostly omitted from this work, but the comparison between updates of the different attention matrices is interesting in its own right so this isn't a great concern

**Questions:**

- Theorem 4.2 is a little hard to parse. Are there any intuitive explanations of $\chi$ and $t$ that could be added?

- It's possible to scale $W_O$ and $W_V$ inversely to each other while not affecting the output of attention. Since it's better to fine-tune the matrix with the smaller norm, does this mean that starting with these two matrices of nearly equivalent norm, we should expect to see an increase in performance by first scaling each matrix so that $W_O$ decreases in norm and then only fine-tuning $W_O$?

- In the toy example, page 3, why do the updates need to be positive?
- Should we expect the norms of the different perturbations to be nearly the same? Are these some experiments to show this?
- Could the second highest element in each Table 1 be highlighted, with an underline? It takes a while to check that K < V < O without it, and with it, that would pop out even better.

- Is there any reason QV was chosen for the experiments in section 4.2, rather than QO or KV?

---

### Official Review · Reviewer_evSn · 2025-11-01

**Soundness:** 2
**Presentation:** 2
**Contribution:** 2
**Rating:** 2
**Confidence:** 4

**Summary:**

This paper provides a theoretical perspective on selecting which blocks to adapt in LoRA-based methods. The authors argue that adapting the Output (O) and Value (V) projection blocks leads to better performance than adapting the Query (Q) and Key (K) blocks, and they establish a connection between these findings and the sensitivity of the softmax function and associated matrix norms. The theoretical claims are validated through experiments on standard image classification and text label prediction tasks.

**Strengths:**

* The paper offers meaningful theoretical insights into why adapting the O and V matrices can be more effective in PEFT.
* The connection between matrix norms, softmax sensitivity, and their influence on adaptation effectiveness is well-motivated and well articulated.

**Weaknesses:**

* The main claim that O and V matrices are the most impactful to adapt in LoRA is not a new empirical observation. As shown in Table 5 of LoRA (Hu et al.), V and O blocks already provide the strongest gains when tuned individually, with O being the most impactful. Thus, while the theoretical explanation is new, the empirical finding itself is not. The paper could benefit from a clearer articulation of how its insights extend beyond those prior observations.
* While the theoretical analysis is valuable, the practical contribution is limited. In most works, all projection matrices are typically adapted with what is likely a small increase in cost compared to full FT, which tends to yield a reasonable trade-off between efficiency and performance.
* The empirical evaluation is limited to relatively simple tasks. Testing the approach on more challenging generative or reasoning tasks (e.g., mathematical reasoning, as explored in DoRA) would strengthen the validation and relevance of the findings.
* The improvements are marginal, and in practice, many PEFT methods (including DoRA) adapt all blocks across both the attention and MLP layers anyway [1, 2] (see Table 11 in DoRA which shows matrices adapted for Llama). Hence, while the theoretical analysis offers useful intuition, its practical impact may be limited. The paper can improve by expanding its analysis to include MLP components and by showing those cost–benefit trade-offs of selective adaptation more explicitly in realistic fine-tuning settings.

Overall, this paper’s main contribution lies in offering a theoretical explanation for a trend already observed (implicitly) in empirical work. The paper can improve by situating its claims better via empirical results on broader practical context (compute efficiency etc) with respect to the practices in SOTA LoRA-based work (tuning almost all blocks with lower rank etc.)

[1] Liu, Weiyang, et al. "Parameter-efficient orthogonal finetuning via butterfly factorization." ICLR 2024. arXiv:2311.06243 (2023).

[2] Liu, Shih-Yang, et al. "Dora: Weight-decomposed low-rank adaptation." Forty-first International Conference on Machine Learning. 2024.

[3] Hu et. al. LoRA: Low Rank Adaptation of Language Models. https://arxiv.org/abs/2106.09685

**Questions:**

* When QV / K blocks are tuned in AdaLoRA/DoRA, do the conclusions regarding the relative importance of O and V blocks still hold? This is not clear from the current experimental setup on those methods. Since these methods achieve substantially larger improvements than LoRA, validating the theoretical claims in those contexts is important.
* It would be helpful if the authors provided a discussion of how their theoretical framework could guide adaptive or data-dependent block selection strategies rather than a fixed choice of O and V blocks. This could make the work more actionable for future methods.

---

### Note · Authors · 2025-11-27

**Comment:**

We authors would like to thank the reviewers for their evaluations, and would further working on the improvements.

**Withdrawal Confirmation:**

I have read and agree with the venue's withdrawal policy on behalf of myself and my co-authors.